# Participatory Rural Appraisal Approaches for Public Participation in EIA: Lessons from South Africa

**Luke A. Sandham** [1,*] **, Jason J. Chabalala** [1] **and Harry H. Spaling** [2]

1   School of Geo and Spatial Sciences, North-West University, Potchefstroom 2521, South Africa;
    jason10368@gmail.com
2   Geography and Environmental Studies, The King's University, Edmonton, AB T6B 2H3, Canada;
    Harry.Spaling@kingsu.ca
*   Correspondence: luke.sandham@gmail.com; Tel.: +27-18-299-1585

**Abstract:** Public participation in environmental impact assessment (EIA) often falls short of the requirements of best practice in the move towards sustainable development, particularly for disadvantaged and marginalized communities. This paper explores the value of a participatory rural appraisal (PRA) approach for improved public participation in a sample of EIA's for photovoltaic projects in South Africa. PRA was conducted *post facto* making use of selected PRA tools. Findings show that a great deal more information was obtained by the PRA approach, confirming the perceived weakness of traditional PP for vulnerable and disadvantaged communities. It is concluded that a PRA approach has considerable potential for improving meaningful public participation, which should improve EIA, build capacity in those communities, and enhance livelihoods and sustainable resource use.

**Keywords:** participatory rural appraisal (PRA); public participation; EIA

## 1. Introduction to EIA in South Africa

Environmental impact assessment (EIA) aims to predict, assess and mitigate changes in the environment attributable to proposed projects before their final approval. After the origin of EIA in the USA in 1970 and its spread through the world, it was practiced voluntarily in South Africa until it became mandatory in 1997 under the Environment Conservation Act (ECA). In 2006 the ECA introduced new EIA regulations promulgated in terms of the National Environmental Management Act (NEMA), followed by further revisions of the regulations in 2010, 2014, and 2017.

The main phases of the EIA process are: Screening and application for authorization, scoping of key impacts, impact assessment to determine significant impacts and suggest alternatives and mitigation, review of the submission for decision-making, environmental authorization, and finally post-decision follow up and monitoring [1]. Since public participation (PP) is regarded as a key attribute of EIA, to allow interested and affected parties to participate in environmental decision making, PP is encouraged in all of these main phases, and ideally should take place as early as possible in the EIA process, but is usually mandated from the time that the application for authorization commences. In South Africa it is mandatory in the application, scoping, and impact assessment phases.

Due to a long history under apartheid of exclusion from decision-making of the bulk of the population until the new democratic dispensation of 1994, the South African EIA system makes provision for comprehensive and extensive public participation, with detailed guidance, and with particularly wide rights to participation to allow maximum inclusiveness.

The prescribed, and widely used methods for PP in EIA usually commence with an invitation to engage, through advertising the process by means of site notices, community letter drops,

press advertisements, and letters to key stakeholders to allow for a register of interested and affected parties to be compiled. Depending on the response to the invitation to engage, public meetings are held, where information documents are distributed, opportunity is given for comments and queries, followed by placement of draft and final reports in public places, and further correspondence by surface mail and email.

Investigation of the effectiveness of EIA has included reviews of the quality of EIA reports (as one aspect of EIA effectiveness) using a quality review protocol, and the report quality reviews have shown that public participation is generally well-performed in terms of what is presented in the EIA reports [1]. However, very little research has investigated the actual effectiveness of PP in EIA, with a particular hiatus regarding PP in vulnerable and disadvantaged rural communities. Studies have shown that EIA PP amongst such communities is not very effective in Ghana [2], for example, and recent research and anecdotal evidence suggest this to be the case in South Africa too [3], most likely because the PP methods in use require a basic level of literacy[1]. Clearly, better methods of public participation need to be explored. Therefore, the aim of this study was to investigate the extent to which a PRA approach might contribute to an increase in the quantity and quality of information gathered during the PP process for EIA's affecting vulnerable communities. We chose the EIA process because it is mandatory, and hence the most regulated and applied environmental management tool, with a large number of EIA's conducted annually.

A number of renewable energy developments are currently taking place in South Africa, mainly wind farms and solar energy installations. In terms of screening requirement, these installations require EIA authorization, including engagement with interested and affected parties through the prescribed PP process. Many of these installations are located on land close to low income residential areas, usually including informal settlements, and comprising of disadvantaged communities. In this case, solar energy is being commercialized as a natural resource. However, the local communities most likely will receive little direct benefit from the utilization of this energy resource. This phenomenon is known as the local resource curse [4]. Rather than the local communities being empowered, they become vulnerable to rent-seeking practices, including land grabbing, corruption, bribery, speculation, and unfair compensation. This increases the imperative for improving public participation by these communities, in order to achieve greater social justice through just mitigation of adverse impacts and more equitable distribution of project benefits. PRA in EIA may be one way of empowering local communities to counter the resource curse.

## 2. Introduction to Participatory Rural Appraisal (PRA)

Participatory rural appraisal (PRA) refers to "a family of approaches and methods to enable rural people to share, enhance, and analyze their knowledge of life and conditions, to plan and to act" [5] (p 953). The scope of PRA now also includes urban residents and other populations such as refugees. PRA emerged during the early 1990s from its predecessor rapid rural appraisal (RRA) and also spawned a later offshoot participatory learning and action (PLA). The origins and distinctions of RRA, PRA, and PLA are described elsewhere [5–10]. This paper uses the term PRA to encompass the broader PRA family.

PRA evolved in response to top-down, institutional, and techno-scientific approaches to development including their econometric and expert-based systems (e.g., field surveys, Geographical Information Systems, modelling) imposed by outsiders. In contrast, PRA advanced using a highly participatory methodology that actively integrated local values and knowledge for decentralized planning and democratic decision making [5,8,11,12], Conceptually, PRA is rooted in

---

[1] South African society is characterized by two distinct components, i.e. developed (mainly white) and developing (mainly black), which reflect the apartheid past. Environmental impact assessment is based in a developed (global north) paradigm, and the uptake amongst the developing community has been relatively weak, partly because of public participation methods that do not adequately address the needs of such communities.

a neo-populist model of development which postulates that community members should be the primary agents of change in development [9,10,13,14]. This model is reflected in "people-centered development", which refers to "a process by which members of a society increase their personal and institutional capacities to mobilize and manage resources to produce sustainable and justly distributed improvements in their quality of life, consistent with their own aspirations" [15] (p. 67). Self-determination, self-reliance, inclusive democracy, social equity, environmental sustainability, and decentralized decision making characterize a people-centered approach. These ideals undergird PRA and embody " . . . participatory ways to empower local and subordinate people, enabling them to express and enhance their knowledge and take action" [8] (p. 3).

Methodologically, PRA is a learning process by which local people gather, analyze, and interpret their own information, and prepare a community action plan based on local values, priorities, and resources. "It is based on the principle that local people are creative and capable and can do their own investigations, analysis, and planning" [12] (p. 3). The process is led by a facilitator trained in PRA but is community-driven to engage local people in sharing their views, knowledge and experiences.

An eclectic toolbox is available for PRA. Tools include semi-structured interviews, transect walks, community mapping, trend and change diagrams, wealth and wellbeing ranking, seasonal calendars, Venn diagrams, causal linkage diagrams, matrix ranking, and scoring [5,6,8,16]. These tools are highly interactive and often carried out in small groups. Many emphasize oral or visual, rather than written, communication to facilitate traditional ways of knowing and inclusion across literacy levels. Certain tools may be associated with specific stages in a PRA process such as community mapping with information gathering, seasonal calendars with trend analysis, and matrix ranking with prioritizing values or planning scenarios. A host of PRA training manuals and handbooks are available e.g., [9,17–21].

PRA applications are numerous and diverse. They represent community planning, market analyses, food security assessment, water and sanitation projects, organizational analysis, policy analysis, and monitoring and evaluation. Many PRA tools have been applied to indigenous environmental knowledge, natural resource management, and climate change, particularly at the local level[2] [16,22–24]. These have enabled a local voice for integrating local environmental values and knowledge into action plans for more sustainable community development [25,26].

Ongoing innovation and adaptation in PRA practice are contributing to further methodological pluralism. For example, PRA is being merged with community-engaged research in the social sciences [27]. PRA techniques are similarly evolving to include new digital technologies such as e-participation [28] and participatory geographic information systems [29].

PRA is not new in South Africa, having been implemented in rural development for some time, particularly post-apartheid in the former homeland areas [30,31]. Enduring lessons from this experience for public participation in EIA are the focus of PRA on human–environment relationships, including indigenous knowledge and values, as well as its potential for enhancing capacity and empowerment among rural communities for sustainable resource development [11,32].

## 3. PRA–EIA Integration

The interactive, participatory nature of PRA would suggest a strong compatibility with the participation component of EIA for several reasons. As a people-centered process, integrating PRA into EIA has considerable potential for meaningful participation that can enhance sustainability learning and action at the local level including for ongoing participatory monitoring and follow up. In developing countries, PRA is often already known among local people because of its wide application in community development and livelihood projects for poverty alleviation. PRA tools emphasize visual

---

[2]   While PRA can be seen as a strategic tool, that does not limit its use in specific contexts and projects, and it therefore offers a powerful method to improve public participation in project level EIA.

and oral communication, which can facilitate understanding of traditional environmental knowledge (e.g., medicinal plants, wildlife habitat, local weather patterns) and enable participation across literacy levels. PRA's benefits of gathering local knowledge in timely, cost-effective ways are also advantages for EIA.

There have been a few calls supporting PRA–EIA integration over the years [33–37] including some practice guidance developed specifically for this purpose [33,38,39]. These calls and guidelines have focused mostly on small development projects such as water supply, agriculture, tertiary roads, and community buildings (schools, clinics), typically implemented by non-governmental organizations. The Calabash Project in Southern Africa has developed a generic framework for public participation in large-project EIA derived, in part, from PRA approaches and methods emphasizing local and traditional knowledge in the region [35,40].

The number of EIAs that have actually applied PRA in any jurisdiction is unknown and only a few case studies of its practice are available. Documented cases include the use of PRA tools (community mapping, transect walks, seasonal calendars, trend lines) for community-based EIA of small agriculture and microfinance (charcoal retail) projects in Uganda [36], demonstration of PRA for scoping valued environmental components and predicting, assessing and mitigating impacts of housing reconstruction in Indonesia [41], and designing and testing of a new PRA tool, a storyboard with photos and captions, for a water project in Tanzania [37].

Overall, application of PRA to the public participation component of EIA is quite limited. Advancing its practice would need to overcome several barriers including:

1. Regulatory and institutional requirements that govern public participation in EIA such as mandated terms of reference, required consultation processes, and fixed timelines, which counter the bottom up, community-driven processes of PRA.
2. Influence and power in EIA tends to be invested in the project proponent, regulatory agencies, review boards or political bodies, and not local people, communities or civil society groups, which PRA advocates.
3. Public participation in EIA is typically managed by outside experts (consultants) with no or little training in PRA facilitation or its tools. On the other hand, trained PRA facilitators have minimal training in EIA and its public participation component.
4. Calls for PRA-EIA integration focus primarily on borrowing PRA tools for their interactive, participatory advantage. For example, community mapping, transect walks, and seasonal calendars are advocated for their participatory utility in EIA. However, integration of a full PRA methodology (process and tools) for meaningful public participation that results in a community-driven sustainability plan in EIA has not been called for to date.
5. Pragmatically, PRA requires trained facilitators, relies extensively on qualitative data, and requires flexible timelines that may not fit well with the professional expertise, quantitative baselines, data-intensive prediction models, and time-bound schedules of many EIA systems.

These barriers have curtailed PRA-EIA integration. Where it has been employed, the nature, extent or type of practice is unknown. No scholarly studies have investigated how PRA has facilitated public participation in EIA in South Africa. Lessons learned for this purpose are timely for filling a gap in the literature and improving future PRA-EIA practice in South Africa and other developing countries facing similar shortcomings of public participation. This paper addresses this need, by focusing on two examples from South Africa.

## 4. Approach

Case study research is defined as "a qualitative approach in which the investigator explores a bounded system (a case) or multiple bounded systems (cases) over time through a detailed, in-depth data collection process involving multiple sources of information ... " [42] (p. 245). A case study represents a research problem that investigates and answers the 'how' and 'why' questions of a

phenomenon within its context [43]. To answer how and why a PRA approach might contribute to an increase in the quantity and quality of information gathered during the PP process for EIA's affecting vulnerable communities, our field research employs a qualitative approach utilizing two case studies representing low income, disadvantaged, and predominantly black communities. Select PRA methods are applied to each case post facto i.e., after completion of the original EIA that used conventional PP methods. The research, in essence, repeats the EIA-PP process albeit independent of any mandatory EIA. Findings from the EIA-PRA process are compared to those of the original EIA-PP process to determine if and how PRA may have changed the quantity and quality of information from community members on potential impacts of the proposed photovoltaic solar farms.

*4.1. Study Area Context*

EIA's were conducted for a series of six photovoltaic (PV) solar farms near to the small rural towns of Theunissen, Koffiefontein, Christiana, Bloemhof, Hertzogville, and the city of Bloemfontein (Figure 1).

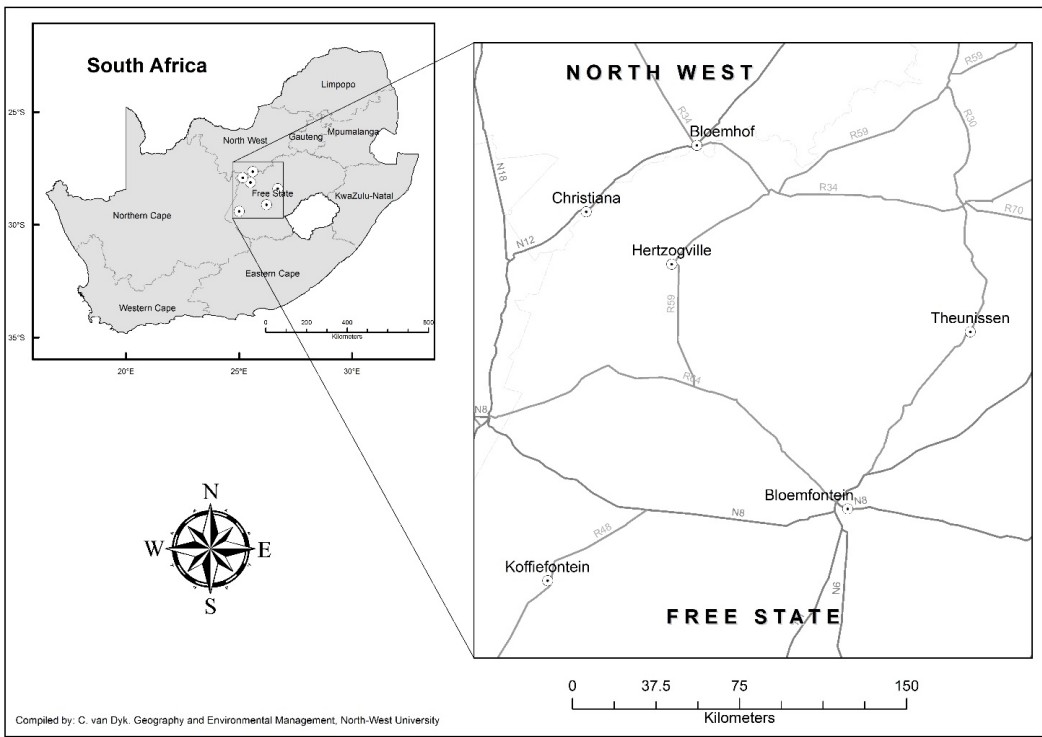

**Figure 1.** Study area.

As part of the EIA process, the officially prescribed public participation process was followed. This entailed the erection of site notices, publication of advertisements in the regional newspapers, letter drops to residents surrounding the sites, and formal notification of statutory consultees, including the local authority and relevant state departments. From this notification process, a register of interested and affected parties was compiled with the goal of inviting their participation in the EIA through PP techniques such as public meetings, letter feedback and the EIA reports made available in the local libraries. Despite the location of the proposed sites of the solar farms relatively close to the low-income townships of these towns, these communities were poorly or not at all represented in the register of parties, suggesting that the prescribed public participation process had failed to engage these low-income, semi-literate communities. For this reason, they were deemed suitable for testing the value of a PRA approach to PP in EIA.

Since the EIA process is linked to strict procedural time frames, and is usually also under pressure for completion from the applicant, it was not possible to implement a PRA approach as part of the

PP process during 2014, and it was therefore decided to conduct a post facto PRA focused on the solar farm EIA's, and the information obtained was compared to the information recorded from the prescribed PP process[3].

*4.2. PRA Engagement in Theunissen and Koffiefontein*

The towns of Theunissen and Koffiefontein were selected for the PRA approach because, of the six communities designated for PV installations, they were the only two with community forums, supporting the respect for community ethos of PRA approach. Community forums are voluntary, non-profit Community Based Organizations (CBOs) formed by individuals in order to promote development (Theunissen) or, as in Koffiefontein, to oppose and redress perceived collusion between the municipality and a large local mine to the detriment of the community. The Theunissen Community forum is no longer active, but at the time of the PRA had about 100 members—mainly white people from the higher income sector. The forum therefore was not representative of the more vulnerable communities for our PRA study, i.e., the community of the low-income black township called Masilo. However, the forum had one member from the black community who was key to introducing the PRA team to the Masilo community. The Koffiefontein Community Forum is much larger with around 1000 members, mostly from the black community, and is still active.

Initial contact was established with the community forums during the EIA PP process, and the groundwork was laid for the PRAs. Each of Theunissen and Koffiefontein were visited for two days by author Chabalala with a translator and a driver. Several PRA tools were adopted for their potential to support the EIA PP process, i.e., questionnaires, focus group discussions, case studies and stories, and participatory digital mapping.

A similar program was followed in both communities, commencing with a meeting with the community forum or its representative, followed by meetings with other individuals and households for the rest of the two-day period, following a random availability sampling approach to select participants. According to the data saturation principle, and the convergence of responses, two days for each town are regarded as adequate. This also exceeds the time devoted to community input in the EIAs in this study. Due to strongly expressed distrust in open community meetings (the result of public meetings being overwhelmed by political agendas and other reason elaborated below), the focus group discussions had to be modified and took the form of several house to house visits, where the households functioned as a small focus group in Theunissen. In Koffiefontein they were conducted in the library (more detail below in Section 5). The discussions were led by guideline questions, and concluded with a semi-structured interview, making use of questionnaires to probe environmental perceptions, the level of awareness of the EIA process and particularly public participation, including notification, opportunity to respond and the role of the public in the EIA process.

Case studies and stories were used to explain complicated information or concepts, e.g., for the principle of sustainability, the respondents were encouraged to tell stories to give their own views. For example, environmental impact was explained through a story of how Mopani worm harvesting caused a decline in the worm population. Participatory digital mapping was conducted using Google Earth on an electronic tablet. Respondents had opportunity to point out familiar landmarks and features in the area, such as their own homes or local schools, and then shifted to the location of the proposed PV facility.

The responses were subjected to content analysis to draw out themes, which are elaborated on below.

---

[3] The PP was an integral part of the EIA, but the PRA was conducted post facto.

## 5. Analysis and Interpretation

### 5.1. Theunissen

The first day commenced with an interview with the member of the Theunissen community forum who represented the community of Masilo (Figure 2) on the forum. Three issues emerged from this discussion. Firstly, the community forum is aware of the development of the photo-voltaic farm, because the forum supports all activities that could contribute to economic growth in Theunissen, for which the proposed photovoltaic farm held potential, and partly because the local librarian supports the activities of the forum, and therefore kept them informed of the EIA reports placed in the library for comment. Secondly, after the EIA PP process, the national Department of Agriculture had opposed the development citing the agricultural-economic potential of the site designated for the photovoltaic farm, despite assurances and studies to the contrary by the landowner and the EIA practitioner respectively. Both the community forum and the landowner suspect that the opposition from the Agriculture Department was the result of corruption within the municipality. This suspicion points to a third issue, the communication gap between the municipality and the community, relating particularly to information on developments initiated by the municipality, but also to other developments involving the participation of the municipality. Consequently, the community members are not engaged in such developments and only find out after decisions have been made. These were the first indications of a continuing theme of poor relationships between the community and the local authority.

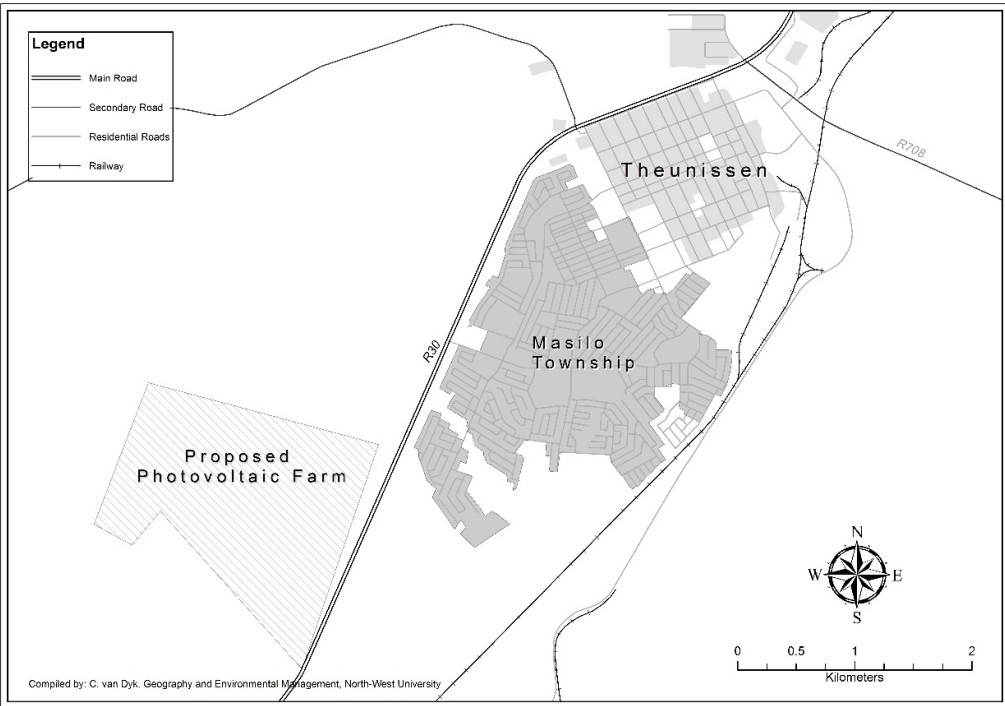

**Figure 2.** Theunissen.

The second day in Theunissen was spent engaging with nine households in Masilo, a low-income settlement virtually adjacent to the proposed photovoltaic farm. Using the random availability sampling approach, households were selected to represent some who were in sight of the PV site, and others further away. On average there were four to seven people in these households, so these discussions involved 45–50 people. Although this engagement originated from the EIA for the photovoltaic development, the issues that emerged were not directly related to the photovoltaic development, but included more generic social issues such as unemployment, sanitation, distrust of

authority, and distrust of public meetings[4]. The research team members themselves experienced some of this distrust, being seen as part of the authorities. Only after the team had carefully explained that they represented neither the local nor provincial authority, nor the developer of the photovoltaic farm, did the respondents demonstrate a degree of relaxation. Thereby they confirmed the view of the community forum member from Day 1, that there is distrust towards (in) the municipality, because of a long history of perceived neglect and ignoring of the needs of the people, leading to a feeling of powerlessness, as stated by one respondent:

> "*I do not see the point in complaining anymore, they do not listen to us. I spoke to my local counsellor about the sanitation situation in my area and he gave me a t-shirt as a form of promise that he would come back to me, needless to say I never heard back from him. So I made peace with the current situation and do not care anymore.*"

There appears to be an expectation that communication from authorities about development proposals should be through verbal announcements during meetings, whereas the notion that the community should consult documents made available in public places is not well accepted. This unmet expectation results in perceptions of the municipality deliberately holding proposals private in order to benefit certain parties, i.e., one respondent:

> "*They (the municipality) don't tell us about any development proposals. They keep the information for themselves, and only share with their family or loyal political supporters.*"

Respondent answers suggested general unfamiliarity about public participation in EIA, apart from one interviewee who works for the municipality. They were unaware that they could raise concerns about the development proposal but commented that even if they did raise their concerns, they did not trust the municipality to consider them.

It became apparent during day two that the main concerns in the Masilo settlement of Theunissen are unemployment, sanitation, political turmoil, and distrust.

The respondents showed little awareness of the value of the environment as a resource, and their views on air, water and soil quality were modulated by a service delivery mind-set. They do have opinions on differences with regard to dislike of littering that again is based on the result of poor service delivery rather than poor environmental quality. This suggested a low level of knowledge of sustainability, but when the concept of sustainability was explained, the respondents indicated that they do value soil quality for better crop production, and good air quality as an indicator of the absence of waste burning.

These findings reinforce anecdotal evidence suggesting that impoverished, vulnerable communities of the global south appear to not have a strong view of the environment as a resource and that their environmental perception is almost entirely controlled by survival and service delivery issues. It also revealed the observed dichotomy in South African EIA where public participation modelled on the global North does not readily involve or engage poor communities [1], and most likely reflects a similar situation in other developing countries where EIA systems based in northern hemisphere environmental paradigms don't fit well into disadvantaged communities [44].

### 5.2. Koffiefontein

One significant difference from the study at Theunissen was that in Koffiefontein (Figure 3) the Community Forum had made a formal appeal against the proposed photo voltaic farm. Following the commencement of the EIA PP process, and the awareness amongst the Community forum about the proposal, a community meeting[5] was held in which it was decided to appeal the proposal because the

---

[4]  The responses presented here for Theunissen and Koffiefontein are a summary of the responses as a whole, reflecting the main themes that emerged from the engagements.

[5]  This was a community forum initiative separate from the EIA PP process.

local authority had not informed the community of the proposal. This hiatus was in contravention of Section 17 of the Municipal Systems Act which requires the local authority to conduct public participation to facilitate engagement of local communities in decision making, all of which *is in addition* to public participation required for the EIA. Additional reasons for the appeal were laid out, including:

- Partnership with the community;
- explanation of choice of a white farmer's land rather than struggling black farmers;
- management of the community trust (a trust created with each photovoltaic facility to transfer some of the income to the local community);
- nature of government contribution and of partners in the venture;
- caution resulting from a previous but failed photovoltaic development on another farm.

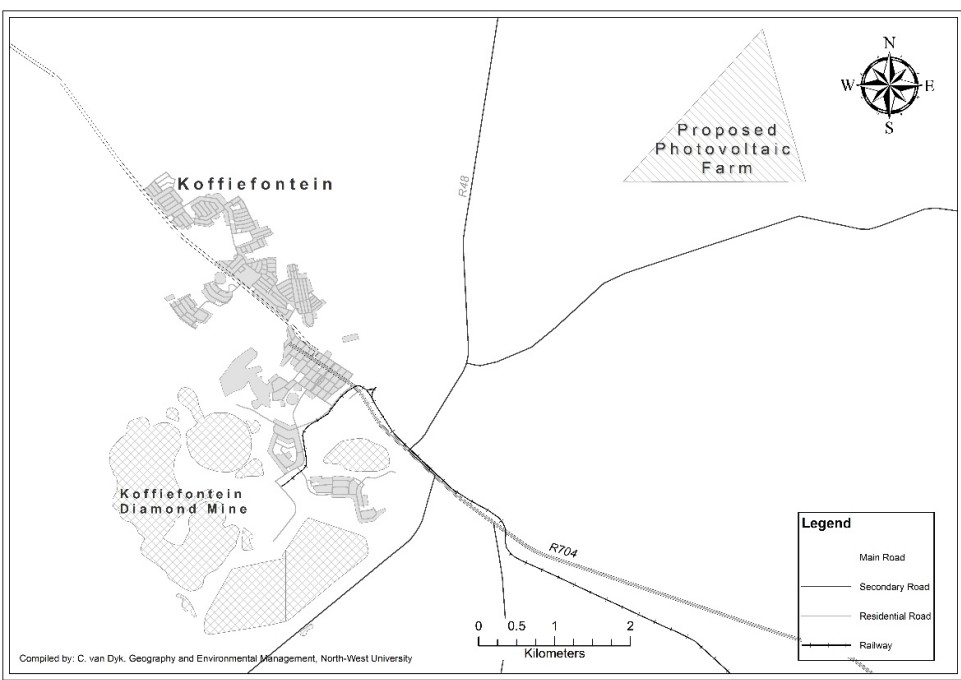

**Figure 3.** Koffiefontein.

The PRA engagement process followed a similar pattern with a meeting on the first day with a member of the Community Forum in the local library, who presented the reasons for the appeal to the PRA team and asked for their responses. Since these are socio-economic matters related to the business plan of the proposed photovoltaic facility, they fell beyond focus of the PRA team which was unable to provide any answers to these questions[6], resulting in a loss of interest in the PRA process by the community forum member. It appears that the loss of interest was carried over from the idea that the photovoltaic farm was being done in secret and with only white farms being selected. Since this forum member was a dominating force in the community, his loss of interest emphasizes a disadvantage of the PRA process, when an important member of the community can obstruct or weaken the PRA. The effect of his withdrawal was that the PRA team did not have the forum's support in connecting with local households and establishing rapport, and was also more vulnerable regarding possible accusations. However, despite these detracting factors, the PRA team was still able to obtain useful information

Day 2 commenced with an interview with the librarian, who was one of the PRA respondents. She demonstrated a high level of awareness of the photovoltaic project, partly because she has been a

---

[6] The authors did not attend the appeal meeting.

municipal employee for a long time and is therefore aware of procedures, and partly because she saw the draft EIA documents that were deposited in the library for public viewing and comment. She did not observe that any members of the community read the EIA documents.

The community forum had advised that respondents be interviewed in the library rather than in their houses and interviews were therefore conducted in the library with a further eight respondents. This had the added advantage that respondents who were working on that day could make time during working hours to come to the library which is a short walk from most parts of town.

The respondents displayed mixed knowledge of EIA PP processes i.e., site notices, press advertisements, and low awareness of the possibility and methods of raising concerns about development proposals through the EIA PP processes. As in Theunissen, the respondents do not ascribe any value to the environment per se, but tend to see it in terms of service delivery e.g., projects in the past provided work, and the environment was clean, in contrast to current unemployment and poor state of the environment.

### 5.3. Comparing Information Gathered during the PP and PRA Processes.

The similarities and differences between the EIA PP for all six[7] solar farms and the PRA processes in Koffiefontein and Theunissen are summarized in Table 1. Data on PP from the original EIA are provided for the six solar farms, including the two at Theunissen and Koffiefontein, to demonstrate that the issues and paucity of information found in these two cases are pervasive among all the communities targeted for photovoltaic installations. For confidentiality purposes, the names of the photovoltaic farms are omitted.

For the EIA PP process, a total of 25 interested and affected parties attended the six meetings, varying between three and five per meeting. The participants were all from the higher income sector including landowners and professionals. The information gathered for all sites was contained in fewer than 10 comments, referring to the following issues:

- A request for a copy of the EIA report;
- impacts on storm water and on surrounding land uses e.g., hunting activities and wildlife;
- security and fire risks, including a landfill; and
- paleontological impacts e.g., artefacts that might be disturbed or destroyed.

The low levels of participation, and sparse information and comments from the EIA PP process are in striking contrast to the outcomes of the PRA process, and these are compared in more detail below.

The demographics of their attendees are the first major differences between PP and PRA (Table 1). The PP process was attended by White landowners (i.e., middle to upper-class citizens) while the PRA was conducted amongst Black and unemployed community members (i.e., disadvantaged people). The differences in environmental values are also visible in the type of information provided through the PP and PRA processes respectively.

The attendees involved in the PP processes value the environment as a resource to be managed to ensure its sustainability, whereas the disadvantaged people involved in the PRA assign a higher priority to the basic human needs like housing, employment and food security. The PRA also showed that the disadvantaged people are not unreachable per se, and they have different issues that take priority for them over environmental related issues. With a slight change in approach, those affected most by developments can be reached and their voices heard.

---

7    EIA PP responses for all six PV farms were compared to show how little information was obtained by that approach.

**Table 1.** Similarities and differences between the public participation (PP) and participatory rural appraisal (PRA) processes for environmental impact assessment (EIA) of six photovoltaic farms in South Africa.

| Criteria | EIA-PP | EIA-PRA |
|---|---|---|
| Number of field days | One day | Two days |
| Number of hours per process | One to two | Two to two and a half |
| Average number of participants | Four | Eight |
| Ethnicity of participants | White | Black |
| Socioeconomic status of participants | Middle-upper class | Lower class |
| Number of communities | Six | Two (of the six for EIA PP) |
| Method of notification | Newspapers, Site notices, letter drops, posted letters | Door-to-door; community venues |
| Mapping tools | Geographical Information Systems, Google Earth | Geographical Information Systems, Google Earth |
| Key informant interviews | Those who know about the PP meeting and can attend are interviewed. | Those not usually involved in PP meetings are interviewed. |
| Community's satisfaction | General satisfaction amongst higher income participants | Little to no satisfaction regarding EIA PP, but more appreciation of the PRA approach |
| Satisfaction with status of environment | Status of environment an important factor. | Status of environment low importance, and only as a result of service delivery |
| Environment as a value | High value and directly proportional to personal wellbeing e.g., healthy environment = healthy yield= high income | Little to no value attached to environment, unemployment a higher priority. |
| Notions of sustainability | Well aware of overall concept of sustainability | Little to no awareness of the concept |
| Important aspects of the environment | Environment is considered important as a whole. | Importance of environment is directly proportional to level of personal gain, e.g., fertile soil is important, but prevention of littering not highly important. |
| Project awareness, including EIA PP | Aware to well aware | Little to no awareness |
| Local knowledge e.g., knowledge about surrounding area/ environmental knowledge | Well informed regarding micro to macro impacts of developments on the environment. | Well aware of local area as a whole, particularly socio-economic dimensions, but no special focus on bio-physical environment. |
| Socio-political context | In opposition to local politicians. | In opposition with local politicians. |
| Governance | Proactively involved in community organizations with the aim of improving what local politics lack. | Passive involvement in community governing attempts, however, community forum is proactively involved. |
| Trust | Trust outsiders/researcher more | Little to no trust of outsiders/researchers |

The PRA approach gathered substantially more information about the local communities that could have influenced decision making regarding these photovoltaic farms. Information such as a community's lack of knowledge regarding EIA and related procedures (social spheres of the environment) are important factors that need to be taken into account when granting Environmental Authorization.

Since environmental management aims to focus on the natural, economic and social spheres of society, it is important that information regarding all three of these spheres is gathered during a PP meeting. Therefore, the fact that landowners were concerned about the rise of crime on their premises and within the community reflect a focus on the social impacts of the proposed photovoltaic farms. Likewise, the PRA revealed that the community as a whole is concerned about unfair use of financial and other resources by municipal officials, which also leads to the rise in crimes like corruption.

Furthermore, the PRA also ascertained that some community members, for example members of the community forum in Theunissen, are concerned about the poor service delivery and misuse of water resources within the municipality. Poor service delivery in particular was an overwhelming theme throughout the PRA process. Although from two different perspectives, both the attendees of the PP and PRA process are concerned about the overall progress and well-being of their communities.

Values and satisfaction related to the environment and notions of sustainability differed among PP and PRA participants. These were rated high and important among white, middle-upper class participants but of little or no importance among black, lower income participants unless the environment related directly to socio-economic gain (service delivery, fertile fields). The situation of different values attached to the environment based on economic stance is not unique to the Theunissen and Koffiefontein communities as it was also observed in Ventersdorp [45].

It was also observed that in both communities the black, low income participants used to attend meetings held by their local municipalities, but after repetitive failure to deliver on promises made by local officials the community started to distrust the municipal system. But officials are not the only deterring factor. For example, the fact that people are unaware that EIA PP processes are advertised in newspapers and site notices might also be a contributing factor. Hypothetically, if someone from the Masilo community manages to see an EIA PP advert, it would still not have made any difference as the person seeing the advert does not know what it is and or what is meant by it.

## 6. Conclusions

The aim of this study was to investigate the extent to which a PRA approach to the current EIA PP system might lead to an increase in the quantity and quality of information gathered during the public participation process. This aim was addressed by conducting post facto PRAs in two disadvantaged communities affected by EIA processes for photovoltaic projects in Koffiefontein and Theunissen. PRA data for these two communities were compared to baseline information gathered from the PP process of the EIA conducted for six communities with photovoltaic projects.

Overall, our findings demonstrate and advance PRA-EIA integration in several ways. The interactive, participatory nature of PRA can facilitate more meaningful participation in marginalized, disadvantaged communities than the rigid information processing tools of PP in conventional EIA. The very nature of PRA approaches will also facilitate engagement of greater numbers of people than the more rigid PP approaches. Building trust and rapport with participants is a major strength of PRA, even in the context of a previous perfunctory EIA that ignored black, low income participants who historically experienced poor communication and political dissonance with local authorities. PRA in these contexts can bring to the fore urgent human needs and priorities with a view to creating sustainable livelihoods and communities as project outcomes.

Regarding its practice, this study showed that PRA tools are adaptable, timely, and cost-effective for meaningful participation in EIA. The Participatory Mapping tool was adapted and changed to an introductory tool, i.e., it was used as a means to give a spatial connection to the proposed developments. Where the interviewees had no idea about the proposed developments at the beginning

of the interviews, they knew exactly where they were situated in real time as the research progressed over the two days, i.e., interviewees became aware of the size and location of the proposed photovoltaic farms. Thus, PRA mapping does not only gather comments and feedback but can also interactively distribute geographic information and increase spatial understanding of a proposed development.

The focus group discussion was adapted to a house-to-house interview. This presented an opportunity to hear and observe how everyone reacts to the same bit of information. The adaptability of PRA facilitated close dialogue in the family home, which turned out to be critical for addressing misperceptions and building trust, which is a condition of meaningful participation. The house-to-house approach also had its own research advantage, as the research team could validate and compare the issues raised during an interview from one house to the other. The case study and storytelling approach needed little adaptation in the field and were useful in conveying information, for example the concepts of sustainable development. Overall, the adaptable and participatory nature of PRA tools are valued attributes for replacing passive and one-way information flows often used in conventional PP.

PRA may also lead to unexpected outcomes. For example, the PRA engaged primarily with members of disadvantaged communities to establish the reason(s) for their absence in the current EIA system. It was discovered that the local political situation played an important role not only in terms of politics but also in areas not directly related to political factions. For example, in Theunissen and Koffiefontein the public has become weary over continued dissatisfaction with service delivery and non-communication issues by their local municipality and refused to continue participating in community-related meetings including EIA PP meetings. The inputs they made in the PRA meeting differed significantly from those made in the EIA PP meetings, all being related to service delivery, unemployment, corruption, and political agendas, with very little on the natural environment. Yet, since these issues impinge on sustainability, PRA cannot only help identify local priority social and economic issues, but also situate these within a normative context of sustainable community development. Furthermore, this research shows that, unlike the information processing tools of EIA PP, PRA can help revitalize local engagement and even be an integrating voice for both sustainability and the marginalized. PRA is rooted in a people-centered, bottom-up approach to rural development focused on self-determination and empowerment. Such an emancipatory view of PRA in EIA assumes that "sustainability cannot happen without the resource needs of local communities becoming a broader economic imperative" [46].

While PRA will not necessarily be able to overcome the gap in communication between local communities and municipal leaders, it can play an important role in terms of the capacity building amongst those most affected by development proposals that are unable to make any contributions due to some disadvantage. Furthermore, methods like Participatory Mapping can also help to bring PP into the digital world in terms of the use of Geographical Information technologies.

When followed properly, a full EIA process from its application to authorization phase can take up to 12 months. With the current EIA system allowing for a major part of the public participation to be conducted before application for authorization is made, PRA can be readily incorporated in such a way that the EIA process does not get prolonged. Furthermore, this research suggests that a PRA approach can be readily incorporated into the scoping and the impact assessment phases where it has a dual potential. Firstly, for improved identification and engagement of stakeholders irrespective of their levels of literacy, thereby increasing the variety and depth of inputs from all sectors of affected communities, as well as contributing improved local knowledge and environmental history. Secondly, it has the potential to add significant value in terms of capacitating disadvantage communities as called for by Simpson and Basta [3]. Moreover, since the disadvantaged communities of Theunissen and Koffiefontein have already been marginalized in the EIA process, they are at risk of becoming victims of the resource curse. Although there is no firm evidence of such rent seeking practices, the information from the PRA suggests that they suspect such behavior, which is not unexpected, since it is a widely occurring phenomenon in many African countries [4]. Therefore, in addition to improving public

participation in EIA, PRA also offers a strong tool to avoid the pitfalls of the resource curse at the community level in South Africa and other developing countries.

**Author Contributions:** Conceptualization, L.A.S.; Formal analysis, J.J.C.; Investigation, J.J.C.; Supervision, L.A.S. and H.H.S.; Writing—original draft, L.A.S.; Writing—review and editing, H.H.S.

**Funding:** This research received no external funding. The APC was funded by North-West University, The King's University and the Social Sciences and Humanities Research Council of Canada

**Acknowledgments:** We thank the people of Koffiefontein and Theunissen that participated in this study

**Conflicts of Interest:** The authors declare no conflict of interest. The funders had no role in the design of the study; in the collection, analyses, or interpretation of data; in the writing of the manuscript, or in the decision to publish the results.

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
