# Peer review of "Participatory Rural Appraisal Approaches for Public Participation in EIA: Lessons from South Africa"

_land, doi:10.3390/land8100150_

Round 1

Reviewer 1 Report

While I feel this is highly rated work, I also have the feeling that there are more questions which should be addressed. I have attached a pdf copy of my comments.

Reviewer 2 Report

Dear authors,

I consider your article to be very interesting and innovative, but in my opinion it needs to be reworked so that even someone who does not have knowledge of the EIA process in South Africa can understand it. Many of your explanations are very general.

It would be useful if you add very short description of EIA process including public participation and methods of public participation in EIA used in South Africa. When you do this, the reader will better understand the contribution of PRA to the issue. It is not clear from your text what are the fundamental differences. The table shows the differences in specific cases, but not in general.

In the discussion, it would be good if you outlined how to proceed in the future and how to involve more people (there are very few interviewed respondents in your case studies to draw conclusions - that I think is another negative of your contribution).

What are main reasons for different involving of "white" and "black" people? Can you explain it? 

My last, but very important, question is focused on PRA environmental connection with EIA process. PRA could be seen as a frame for future projects (and future EIA), but it is linked to SEA and public participation during SEA process. So, my question is: why did you chose the EIA process?

Reviewer 3 Report

The authors bring forth a very interesting topic on how to bridge PRA and EIA.

The introduction gives the reader a good overview of the topic. Towards the end of this section, the authors should introduce the aims and objectives of this research. Currently it lacks on clarity to what is being explored.

Section 2, PRA-EIA integration, this section lacks on clarity around EIA. Why is EIA important? How does it complement or not PRA? Why other approaches aren't considered?

Section 3, EIA in South Africa, provides a good overview of the major events and developments in South Africa. I'm wondering if we would need to develop a parallel argument between EIA and PRA. Currently it feels very disconnected.

I also think a clear methodology sections should be included in this paper. I can see parts of it across the previous sections but is not clear how the authors decided for a specific methodology. It is not clear as well why a case-study approach and what criteria has been applied to select such case studies.

Section 4, more information is needed in terms of how those workshops and interviews have been carried. What sort of format was applied? how many people did attend? what questions were asked? what information was collected? how that data was analysed?

Section 5 is the strongest in this paper. the authors described the results very objectively and drawn out important conclusions.

Section 6, conclusion, is well-written and summarises the study very clearly.

Another important point is that the authors do not mention any ethical approval for this study. Clarity is needed here.

Round 2

Reviewer 1 Report

I think the authors have endeavoured to address most of the concerns I raised. While other areas were not addressed, they gave a strong rebuttals which is acceptable in academia.

I therefore suggest that their paper is accepted for publication in its current form pending the editors judgement and I  wish them the best.

Author Response

Thank you for the positive response to our first round of revisions.

No further response required.

Reviewer 2 Report

Dear authors,

thanks for your revision of the manuscript, you did a good work. But still I have a comment on the description of the EIA and PP process. Yes, you have added some information about the EIA in South Africa, but you need to add it. What are main steps of EIA in South Africa and in which step of EIA is PP mandatory? Could you create a scheme which could clarify it?

Yes, you are right, you have used PRA as a method to improve PP in EIA, my point was that PRA is a strategic document, so SEA methods are used. 

Author Response

Comments:

1.  But still I have a comment on the description of the EIA and PP process. Yes, you have added some information about the EIA in South Africa, but you need to add it. What are main steps of EIA in South Africa and in which step of EIA is PP mandatory? Could you create a scheme which could clarify it?

2.  Yes, you are right, you have used PRA as a method to improve PP in EIA, my point was that PRA is a strategic document, so SEA methods are used.

Response:

1.  Thank you.  These have been addressed through additions in the introduction. (Line 25 forward).

2.  Thank you. To address this, we have included a footnote to recognize its strategic base along with utility at local level, where the local use of PRA is cited in the manuscript at Lines 117-120. 

“Many PRA tools have been applied to indigenous environmental knowledge, natural resource management, and climate change, particularly at the local level [16][22] [23][24]. These have enabled a local voice for integrating local environmental values and knowledge into action plans for more sustainable community development [25][26].”

See also lines 125-130 and cited references.